# Anthelmintic Efficacy and Pharmacokinetics of Ivermectin Paste after Oral Administration in Mules Infected by Cyathostomins

**DOI:** 10.3390/ani10060934

**Published:** 2020-05-28

**Authors:** Marilena Bazzano, Alessandra Di Salvo, Manuela Diaferia, Fabrizia Veronesi, Roberta Galarini, Fabiola Paoletti, Beniamino Tesei, Amy McLean, Vincenzo Veneziano, Fulvio Laus

**Affiliations:** 1School of Biosciences and Veterinary Medicine, University of Camerino, Via Circonvallazione, 62024 Matelica (MC), Italy; marilena.bazzano@unicam.it (M.B.); beniamino.tesei@unicam.it (B.T.); 2Department of Veterinary Medicine, University of Perugia. Via S. Costanzo, 06126 Perugia, Italy; alessandra.disalvo@unipg.it (A.D.S.); manuela.diaferia@unipg.it (M.D.); fabrizia.veronesi@unipg.it (F.V.); 3Istituto Zooprofilattico Sperimentale dell’Umbria e delle Marche “Togo Rosati”, Via Salvemini, 06126 Perugia, Italy; r.galarini@izsum.it (R.G.); f.paoletti@izsum.it (F.P.); 4Department of Animal Science, University of California Davis, Davis, CA 95616, USA; acmclean@ucdavis.edu; 5Department of Veterinary Medicine and Animal Productions, University of Naples Federico II, 80137 Naples, Italy; vinvene@unina.it

**Keywords:** ivermectin, mule, cyathostomins, pharmacokinetics, anthelmintic efficacy

## Abstract

**Simple Summary:**

Mules and donkeys are often treated as horses from a therapeutic point of view. This approach could be dangerous due to species differences in drug pharmacokinetics which could reflect on the drug effectiveness. Ivermectin is a commonly used anthelmintic compound due to the broad spectrum of activity. The improper use of ivermectin (i.e., dosage, route of administration) could cause a lack of parasite control and contribute to development of drug resistance. Studies on the pharmacokinetics and efficacy of antiparasitic molecules in mules are limited, although these drugs are crucial for the welfare of these equines. The aim of the present study was to evaluate the efficacy and pharmacokinetics of ivermectin administered to mules at the same dosage (200 µg/kg body weight) and route licensed for horses. Results show that administering ivermectin orally, at the same dosage of horses, has a pharmacokinetic intermediate behavior between horses and donkeys. This study demonstrates that ivermectin oral paste at horse dosage is effective and safe for the treatment of cyathostomins in mules.

**Abstract:**

Ivermectin (IVM) is an anthelmintic compound commonly used off-label in mules due to its broad-spectrum of activity. Despite the general use of IVM in mules with the same dose and route of administration licensed for horses, significant pharmacokinetic differences might exist between horses and mules, as already observed for donkeys. The aim of the present study was to evaluate the pharmacokinetic profile and anthelmintic efficacy of an oral paste of IVM in mules naturally infected with cyathostomins. Fifteen adult mules with fecal egg counts (FEC) ≥ 200 eggs per gram (EPG), with exclusive presence of cyathostomins, were included in the study. All mules were orally treated with IVM according to the manufacturer's recommended horse dosage (200 µg/kg body weight). FECs were performed before (day-10 and day-3) and after treatment at days 14 and 28 by using a modified McMaster method. The FEC reduction (FECR%) was also calculated. Blood samples were collected from five animals at various times between 0.5 h up to 30 days post treatment to determine pharmacokinetic parameters. The maximum IVM serum concentration (Cmax) was 42.31 ± 10.20 ng/mL and was achieved at 16.80 ± 9.96 h post-treatment (Tmax), area under the curve (AUC) was 135.56 ± 43.71 ng × day/mL. FECR% remained high (>95%) until the 28th day.

## 1. Introduction

The estimated global equine population is 117 million, including 58 million horses, 50 million donkeys and 9 million mules [1]. Although there has been a decrease in mule populations in Europe and the Mediterranean European countries, mules are still widely used as a beast of burden or working equids for draft purposes in developing countries because they are considered to be strong and hardy animals [1,2]. Furthermore, in some countries such as the U.S. there has been a growing interest in using mules also for recreational riding, racing and show purposes [3,4]. The increase in mule use has created the need for more information on how to properly treat and care for them [4].

Little research is available for donkeys but even less is known about mules [5]. In literature, only limited data is available on the dosage, pharmacokinetics and efficacy of drugs in mules [6,7]. The lack of scientific information for mules may be due to the idea that these equids are considered less demanding from a health point of view (mules are considered less susceptible to diseases and fatigue compared to horses) or the fact that they are mostly reared in parts of the world where little routine veterinary care is available [8]. However, mules may be susceptible to the same parasites affecting horses and donkeys. Among all nematodes that may have the greatest negative impact on the wellness of mules, cyathostomins (small strongyles) are of major concern and efficient control based on the appropriate use of anthelmintic drugs is mandatory [9,10].

The three anthelmintic drug classes (e.g., benzimidazoles, pyrimidines and macrocyclic lactones) [11] registered for horses are currently not available for treatment of parasites in mules and only few drugs are specifically licensed for use in other equids like donkeys [12,13,14]. Therefore, mules and donkeys are usually treated with anthelmintic drugs at the same dosage, route and intervals licensed for horses, despite the lack of scientific reports evaluating pharmacokinetics and efficacy related to their use in these animals [11,15,16].

Among anthelmintic compounds registered for equine species, ivermectin (IVM) is a macrocyclic lactone (ML), commonly used due to its broad-spectrum of activity against both endo- and ectoparasites [11,17]. Ivermectin administered as oral paste, at dosage of 200 µg/kg body weight (BW), is characterized by a highly safe margin and extensively used in the equine industry [18,19,20]. Ivermectin is a very lipophilic drug and, in general, its pharmacokinetics in domestic species is characterized by a slow phase of absorption, extensive distribution, scant metabolization and slow excretion, principally with feces [21]. In horse species the oral administration of IVM shows a faster absorption compared to subcutaneous injection [22], a mean resident time in the organism that ranges from about four to seven days [20,23,24] and a slow elimination with feces in which the maximum concentration is achieved after two days from administration [25].

Despite the general use of IVM in the mule at the same dose and route indicated for horses, significant pharmacokinetic differences might exist between horses and mules, as already observed between horses and donkeys both for IVM [26] and other drugs [27,28]. Some differences in pharmacokinetics and efficacy have already been observed between mules and horses for antibiotics (i.e., sulfamethoxazole and trimethoprim) [6] and for sedatives (i.e., xylazine) [7]. 

Anthelmintic activity depends on the interaction between the active drug ingredient and the specific receptors in the parasitic target, but also on drug concentrations obtained at the parasitic site [29,30]. Animal species is one of the main factors affecting the blood drug disposition and thus the concentrations attained for the parasitic targets.

The aim of the present study was to evaluate the blood disposition and the anthelmintic efficacy of an oral formulation of IVM administrated in mules naturally infected by cyathostomins, in terms of percentage of fecal egg count reduction (FECR%), following the international guidelines recommended by the American Association of Equine Practitioners (AAEP) [31].

## 2. Materials and Methods

### 2.1. Animals and Experimental Groups

Fifteen adult mules, ten females and five males, with a mean age (±standard deviation) of 11 ± 6 years and a mean weight (±standard deviation) of 511 ± 48 kg were included in the study. Weight was estimated as previously described [32]. Owner’s consent was given for all animals included in the study. 

The mules were reared in central Italy as working animals and were kept in outdoor paddocks with no shared pasture with other equids. The management was the same for all the mules, animals were allowed to graze for several hours per day and their diet consisted of fresh forage (grass), 8 kg grass hay and 4 kg concentrates per day, and water ad libitum. All the animals enrolled in the study belonged to the same farm and shared the same paddocks from at least 12 months. The mules received treatments for gastrointestinal parasites occasionally in previous years, and no anthelmintic drugs were administrated to the mules in the previous 12 months. 

Before starting the study on day −10 (D_−10_), clinical examinations and hematological investigations were performed on each animal to ensure their health status. Parasitological examinations, consisting of individual fecal egg count (FEC) followed by coproculture, were performed at ten (D_−10_) and three (D_−3_) days before anthelmintic administration. All mules used in the study had a FEC ≥ 200 eggs per gram (EPG) with exclusive presence of cyathostomins. A total of fifteen mules were tested for IVM efficacy, in accordance with the guidelines of the World Association for the Advancement of Veterinary Parasitology (WAAVP) [33].

The study was conducted with the favorable consent of the local organization for animal welfare (OPBA, protocol number E81AC.10) of the University of Camerino and the approval of Ministry of Health (protocol number 677/2018-PR) in accordance with directive 2010/63/EU of the European Parliament on the protection of animals used for scientific purpose.

### 2.2. Treatment Procedures

On day 0 (D_0_), a commercially available equine oral paste formulation of IVM (Eqvalan Oral Paste, Merial Italia, Boehringer Ingelheim Animal Health) was administered orally to each mule six hours before feeding (after a fasting period of 12 h) at the horse dosage of 200 µg/kg of BW. All treated animals were observed continuously for the first three hours after drug administration and then weekly until the end of the trial period to detect possible adverse reactions following treatment.

### 2.3. Sampling Procedures 

Individual fecal samples were collected from the rectum of each animal on D_−10_, D_−3_ and D_0_ immediately prior to anthelmintic administration, and then on day 14 (D_+14_) and day 28 (D_+28_) post treatment. According to general recommendations proposed by Nielsen et al. (2010) [34], the fecal samples were stored at +4 °C waiting for FEC.

In order to assess IVM pharmacokinetics 5 of the 15 treated mules (females, between 6–21 years), chosen on the basis of a mildest character, and consequently more easily manageable, were enrolled. Whole blood samples (about 10 mL) were collected by jugular vein puncture before drug administration and then at predetermined time-points: 30 minutes (min), 1, 2, 4, 6, 8, 10 and 24 hours (h) and 2, 3, 4, 7, 10, 15, 20, 25 and 30 days (d) after the treatment. Blood samples were centrifuged at 2500 g for 10 min and the serum was collected and stored at −80 °C until estimation of drug concentration.

The sample size as well as the timing of blood sampling was assessed on the basis of previous studies on IVM pharmacokinetics in equine species [20,22,23,24,26]. 

### 2.4. Coprological Examinations

Individual FECs were performed in all mules at D_−10_, D_−3_, D_0_, D_+14_ and D_+28_, using a modified McMaster technique with a lower detection limit of 50 eggs per gram (EPG) and a saturated sodium chloride (NaCl) solution with a specific gravity of 1.200 [35]. Pooled fecal cultures were performed at D_−10_ and D_−3_, while individual fecal cultures were performed for all mules with FEC ≥ 200 EPG at D_0_, D_+14_ and D_+28_. Fecal cultures were incubated at 27 °C for 7–10 days for larval development and 3rd stage larvae were identified according with morphological identification keys [35]. When a coproculture had 100 or less third stage larvae, all were identified; when a coproculture had more than 100 larvae, only 100 larvae were identified.

### 2.5. Analytical Determination of Ivermectin

One milliliter of serum was added to a 15 mL Falcon tube with 50 µL of a methanolic solution of the internal standard, ivermectin, at 1 µg/mL, 2.5 mL of acetonitrile and 4 g of anhydrous sodium sulphate. After centrifugation, the supernatant was collected in a new tube and the extraction was repeated with 2.5 mL of acetonitrile. The combined extracts were evaporated to dryness and then re-dissolved in 200 µL of water and 2.5 mL of ethyl acetate. The samples were purified through a SPE cartridge (NH2, 100 mg/3 mL, Biotage IST Isolute, Uppsala, Sweden) previously conditioned with 3 mL of methanol and 3 mL of ethyl acetate. Furthermore, 3 mL of ethyl acetate was used to wash the sample tube and transfer onto the SPE cartridge. The eluate was evaporated until dry and the analyte derivatization was executed according to a previously published method [36]. After derivatization, the samples were injected into a high-performance liquid chromatography (HPLC) system. 

The HPLC equipment consisted of a Thermo Finnigan Spectrasystem (San Jose, CA, USA) with a P4000 quaternary pump, an AS3000 autosampler and a fluorescence detector (FL3000). The separation was achieved on a Luna C8 (150 × 3.0 mm, 5 µm, Phenomenex, Torrance CA, USA) analytical column equipped with a guard column C8 4 × 2.0 mm (Phenomenex). The mobile phases were acetonitrile (A) and water (B). The flow rate was 1.2 mL/min and the injection volume 50 µL. The gradient profile was as follows (1) 0–3 min, to 85% A; (2) 3–7 min, to 97% A; (3) 7–10 min, 97% A; (4) 10–12 min, to 85% A; and (5) 12–14 min, to 85% A. The total run time per single injection was 14 min. Fluorescence of the derivatized compounds was detected at excitation and emission wavelengths of 364 and 470 nm, respectively. In each analytical batch, a series of five concentration points (2.5, 5, 25, 50 and 100 ng/mL) prepared by spiking blank serum samples were injected for calibration. 

Validation parameters were determined in accordance with EMA guidelines on bioanalytical method validation [37]. Five spiked samples were analyzed at four concentrations (5, 50, 100 and 200 ng/mL) on two different days. The observed recoveries ranged from 90% to 106%. Within-run and between-run precisions were in the range 3.5–8.5% and 8.5–13%, respectively. The lower limit of quantification (LLOQ) was 2.5 ng/mL, whereas the upper limit (ULOQ) was 200 ng/mL.

### 2.6. Pharmacokinetic and Anthelmintic Efficacy 

Pharmacokinetic parameters were calculated, for each animal, from serum concentration–time data executing non-compartmental analyses by PK-Solver program [38].

To determine the efficacy of IVM against cyathostomins at each fecal sampling time, an arithmetic mean of EPG was calculated following the American Association of Equine Practitioners (AAEP) parasite control guidelines [31]. For each mule, the percentage efficacy (%) was assessed in terms of fecal egg count reduction (FECR) at different days (D_+14_ and D_+28_) using the following formula:

FECR (%) = 100 × (Mean EPG pre-treatment − Mean EPG post-treatment/Mean EPG pre-treatment).

### 2.7. Statistical Analyses

Microsoft Office Excel 2016 software was used for data recording, and FECR, expressed as percentage with 95% confidence intervals, was calculated using the RESO FECRT analysis program, version 4 (http://sydney.edu.au/vetscience/sheepwormcontrol/). The values of the FECR were interpreted according to the AAEP parasite control guidelines [31]. The 95% lower confidence limits (LCL; %) observed for all time points was selected so that resistance would be indicated if the % mean FECR was below 95% and the LCL was below 90% [39].

## 3. Results

### 3.1. Parasitological Results

Clinically, none of the animals enrolled in the current study showed any adverse reaction following IVM administration.

The FECs (mean, range and standard deviation) obtained at the different sampling times and the results of the FECR with 95% CI are shown in Table 1. At baseline (D_0_), mules included in the study had an average of 1360 EPG, consisting of 100% of cyathostomins; the mean EPG at D_+14_ and D_+28_ were 46.67 and 26.67 EPG, respectively and did not differ statistically (*p* > 0.05) (Appendix A). The FECR was 96.57% and 98.04% at D_+14_ and D_+28_, respectively. The lower 95% CI was over 90% at D_+14_ and D_+28_.

### 3.2. Pharmacokinetics Results

Ivermectin was quantifiable in serum 30 min following drug administration in two mules, after 1 h in two mules and after 2 h in one mule. The concentrations of IVM in systemic circulation ranged from 4.2 to 7.4 ng/mL at the first time of appearance in systemic circulation, achieving the Cmax (42.31 ± 10.20 mean ± SD; range 28.3–56.7) between 4 h (only one mule) and 24 h post-treatment. Finally, the serum concentration of IVM decreased progressively, remaining detectable in all animals up to seven days post administration. In one animal, the drug was detectable in serum for 10 days post oral administration, 15 days in another mule and until 20 days in one mule. Figure 1 illustrates the trend of serum IVM concentrations in relation to time. Table 2 shows the main pharmacokinetic parameters of IVM in serum following *per os* administration at dosage of 200 µg/kg BW following non-compartmental analysis. 

## 4. Discussion

The current study represents the first study that has explored both pharmacokinetics and efficacy of IVM orally administered in mules. Ivermectin orally administered at 200 µg/kg BW showed intermediate pharmacokinetic parameters between horses and donkeys and seems to be efficacious against cyathostomins.

Ivermectin was the first ML approved in horses as a broad spectrum antiparasitic compound at the recommended dose of 200 µg/kg BW [18]. Since its introduction as an antiparasitic drug in equid species, many authors have investigated the efficacy of IVM on different parasites when administered orally to horses, and also its pharmacokinetics has been well documented [20,22,23,24,40]. However, there is a paucity of data available in the literature on the pharmacokinetics and antiparasitic efficacy in other equids. Even if donkeys, mules and horses are morphologically similar, they can have considerable differences in pharmacokinetic profiles [6,41,42]. Several studies report a greater capability of drug elimination in donkeys than in horses [13,43,44]. Latzel et al. (2012) [7] observed a significant difference in the half-life of xylazine between horses and mules following IV administration, and at the same time, a less intense sedation in mules. A mule may require up to 50% more of a sedative when compared to a horse to obtain an appropriate level of sedation [45]. Thus, the importance of studying the pharmacokinetics of each drug in the specific target species is evident.

The value of efficacy obtained at D_+14_ in the present study may be considered correctly estimated according to the recommendations by Kaplan and Nielsen (2010) [46], in fact FECR tests have been calculated in fecal samples collected 14 days after treatment and not in the early period post-treatment when ML might cause temporary suppression of the egg shedding and thus leading to a overestimation of the anthelmintic efficacy. However, the obtained FECR values showed an efficacy rate of IVM against cyathostomins lower than those detected in previous studies conducted in horses and donkeys ranging from 96% to 100% [10,47,48,49]. Since a FECR% ranging between 95–98% should be considered suspected of resistance according to the AAEP guidelines [31], we decided to perform an additional FEC at Day_+28_ to assess if a higher decrease in efficacy may support suspected anthelmintic resistance. ML are characterized by very long ERPs (egg reappearance periods), but recent reports have documented ERPs shortened to 4–5 weeks which may be considered an early indicator of resistance to MLs [31]. The FECR obtained at D_+28_ (98.04%) was higher than D_+14_ (96.57%), which is different from what we would expect; however, the average FECs obtained at two distinct sampling times did not differ statistically. Possible reasons for why an increased percentage in the FECR at D_+28_ was observed, could be related to: (i) few individual variations of egg excretion commonly observed in horses and possibly in mules, (ii) scant efficacy of IVM on the encysted parasites (L3–L4 larval stages) that could conclude their cycle within 28 days increasing the FEC post-treatment and (iii) the analytic sensitivities of the McMaster technique based on a conversion rate, which is however still considered the reference method for the FECR tests. 

Consideration for resistances to IVM remained inconclusive because we detected borderline reduced efficacy rates at D_+14_ and D_+28_. Unfortunately, we were not able to continue to calculate ERP due to behavioral resistances from the animals making it impossible to continue the collection process.

However, borderline reduced efficacy detected in the present study could be related to other factors not associated with resistance. For example, the concern that the sub-optimal dosing of 200 µg/kg of BW may relate to reduced efficacy. We think that it is extremely unlikely that resistant cyathostomin populations infected the sampled mules for a number of reasons: (i) the animals did not share pastures with other equids e.g., horses or donkeys, that could harbor resistant strongyle strains; (ii) any external introduction of animals was done in the last 10 years and (iii) pharmacological pressure within the herd was minimal since the anthelmintic treatments were only occasional.

In this study, the Cmax and AUC observed in mules (42.31 ± 10.20 ng/mL and 135.56 ± 43.71 ng × day/mL, respectively), when treated orally with 200 µg/kg of IVM, are greater than that observed in donkeys (23.4 ± 4.4 ng/mL and 119.3 ± 12.3 ng × day/mL) by Gokbulut et al. (2005) [26] and similar for horses (44.0 ± 23.1 ng/mL and 132.7 ± 47.3 ng × day/mL) [23]. 

Likewise, Coakley et al., (1999) [50], following administration of flunixin in mules, observed a greater similarity to horses rather than donkeys in drug serum levels.

Nevertheless, it is important to underline that in other studies, after oral IVM administration to horses, greater Cmax and AUCs were seen in respect to the present study [22,24]. The reasons for the differences among these various studies are not clear, but could be due to a different feeding regime. Some authors have speculated that IVM could be absorbed by the particles of digesta causing a low availability of the drug after its oral administration [20,22]. In order to maximize standardization of the treatment conditions in this study, we used a protocol that included fasting for 12 h followed by feed administration 6 h after treatment. Other factors that could lead to a difference in results in above-cited studies could be differences in environment, age, gender and level of fitness as already hypothesized for hematological and blood biochemistry among different equids [4]. Another thought on inconsistent results could be related to a different expression of P-glycoprotein (which has been shown to be involved in the absorption of IVM following its oral administration [21]) in intestinal epithelium of mules compared to other equids. In addition, because IVM is a very lipophilic drug that accumulates in adipose tissue, a different body fat content in mules, compared to horses and donkeys, could be responsible for a different drug distribution and, consequently, for a different profile of drug serum concentrations [21]. Lastly, the different LOQ of the analytical method could have affected the computation of the AUCs [51].

In this study, the Cmax was achieved later than in horses (16.8 h vs. a range, expressed as mean value, between 3.3 and 9.2 h) [20,23,24] and before than in donkeys (24 h) [26]. In three out of five mules, the Tmax was obtained at 24 h; this value was closer to that observed in donkeys compared to that seen in horses [26]. It is important to highlight that the large individual variability observed in serum drug concentrations in the present study, did not allow for generalization of pharmacokinetic behavior of IVM in mules being closer to horses or donkeys. The same large individual variability in systemic drug concentrations was observed in other pharmacokinetic studies on IVM [23,24].

Unfortunately, according to our knowledge and review of literature, there are no PK/PD predictors of efficacy for anthelmintic agents, contrary to antibacterial drugs. Therefore, currently it is not possible to evaluate the potential clinical impact of IVM, orally administered at doses of 200 µg/kg BW in mules, on other parasites. This study may be considered as a first step towards knowledge about pharmacokinetic properties of IVM in mules; further studies are warranted in the future.

## 5. Conclusions

Studies on correct use of anthelmintic molecules in mules could contribute to improved control of parasites and, consequently a gradual increase in their welfare, supporting a rising of the population worldwide. The present study demonstrated that IVM administered orally at 200 µg/kg BW to mules has intermediate pharmacokinetic parameters between horses and donkeys. Furthermore, IVM seems to be efficacious against cyathostomins in mules even if an optimization of the dosage should be obtained based on further studies. 

## Figures and Tables

**Figure 1 animals-10-00934-f001:**
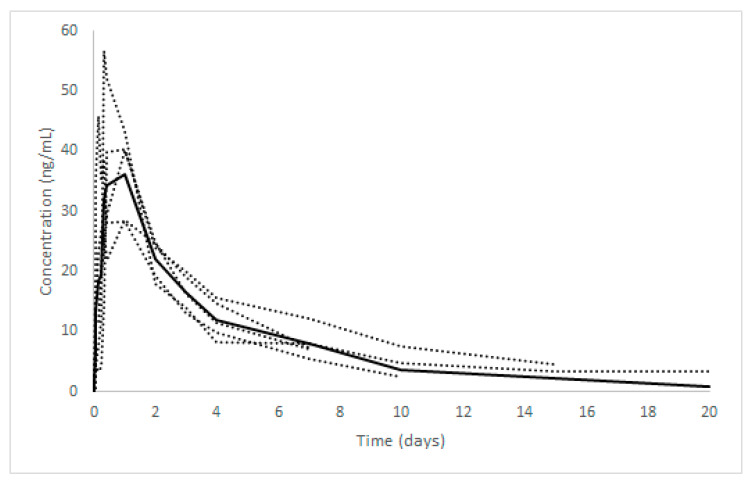
Average (solid line) and single (dot lines) IVM serum concentrations vs. time after oral administration at a dose of 200 µg/kg BW in mules (*n* = 5).

**Table 1 animals-10-00934-t001:** Initial (Day_0_) and post-treatment (D_+14_ and D_+28_) fecal egg counts (FECs) expressed as mean of egg per gram (EPG), relative standard deviation (SD) and percentage of fecal eggs count reduction (FECR%), with their 95% lower and upper confidence intervals (CIs) after an oral treatment of ivermectin (IVM) in mules naturally infected by cyathostomins.

Days	FECs EPG AM	EPG Range	SD	FECR (%)	Lower 95% CI	Upper 95% CI
0	1360.00	200–3550	908.46	-	-	-
14	46.67	0–250	68.26	96.57	90.65	98.74
28	26.67	0–200	57.06	98.04	93.40	99.42

Abbreviations: FECs, fecal egg counts; EPG, egg per gram; AM, arithmetic mean; SD, standard deviation; FECR, fecal egg count reduction; CI, confidence interval.

**Table 2 animals-10-00934-t002:** Main pharmacokinetic parameters of ivermectin following oral administration at 200 µg/kg BW in mule (present study), horse and donkey (derived from previous published studies).

Species (Number of Animals)	Parameter (Unit)						Reference
	t_1/2__λ__z_	T_max_	C_max_	AUC _0–t_	AUC _0–∞_	MRT	
	(Day)	(h)	(ng/mL)	(ng × Day/mL)	(ng × Day/mL)	(Day)	
Mule (*n* = 5)	2.74 ^§^ ± 2.02 ^†^	16.8 ± 9.96	42.31 ± 10.20	135.56 ± 43.71	163.93 ± 61.82	6.07 ± 4.36	present study
Horse (*n* = 6)	6.53 ± 0.92	4.08 ± 2.16	61.28 ± 10.73	164.96 ± 30.07	-	7.34 ± 1.30	Gokbulut et al., (2010) [24]
Horse (*n* = 5)	4.25 ^§^ ± 0.24 (a)	9.22 ± 5.71	44.0 ± 23.1	132.7 ± 47.3	-	4.78 ± 0.64	Perez et al., (1999) [23]
Horse (*n* = 5)	2.93 ± 0.4 (a)	3.60 ± 0.96	51.3 ± 6.1	137.1 ± 35.9	-	4.2 ± 0.4	Perez et al., (2003) [20]
Horse (*n* = 3)	2.76 ± 0.2	3.3 ±0.7	82.3 ± 12.4	200.92 ± 22.67	-	-	Marriner et al., (1987) [22]
Donkey (*n* = 3)	7.4 ± 2	24.0 ± 0.0	23.6 ± 4.4	119.3 ± 12.3	-	6.5 ± 0.2	Gokbulut et al., (2005) [26]

(a): terminal half-life resulted from triexponential equation; AUC _0–t_: area under serum concentration–time curve from zero up to the last concentration ≥ LOQ; AUC _0–∞_: area under serum concentration–time curve from time zero to infinity; C_max_: maximum concentration observed; MRT: mean residence time; T_max_: time of maximum concentration observed (when expressed in days in published studies, it was converted to hours according the formula h = day value × 24); t_½__λz_: terminal half-life; SD: standard deviation; ^§^ harmonic mean; ^†^ pseudo standard deviation.

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
