# Peer review of "Anthelmintic Efficacy and Pharmacokinetics of Ivermectin Paste after Oral Administration in Mules Infected by Cyathostomins"

_animals, 2020, doi:10.3390/ani10060934_

Round 1

Reviewer 1 Report

A well designed study considering the differences between equid species an areas that has currently seen activity in the microbiome field.

Clear abstract and rationale with aim - was there a hypothesis at the outset?

Methods section is generally clear- where the sampling times for pharmacokinetics based on the previous horse studies? A comment to this effect would be useful.

Good conclusions on the findings

Main areas for revision:

The discussion section requires some work in the English language and also further clarity and highlighting limitation which currently are lacking.

To clarify were all the mules on exactly the same management and diet?

Why were only females used for blood sampling and why only 5? Is there rationale for this?What effect might this have on statistical power? Worth a limiitation comment in the discussion. Similarly why 15 mules how might the sample size reflect findings and how comparable is this to the other horse studies cited? 

Line 232 - comment on suppression of egg laying activity - do you have a citation for this - we have seen it in horses for Benzimidazoles

Line 235- comment on ERP - ERP was calculated at 28 days but not further. The guidelines used here AAEP are good but in the literature there are multiple ways to calculate ERP e.g. Relf et al 2014 and Daniels & Proudman 2016 - both used 90% reduction post 14 days as reduced efficacy for which you did not observe, therefore you observed good efficacy in this study. 

It might be of value to comment on the encysted burden? This is possible to test via serum from Austin Davies Diagnostics - could this have any influence on findings?

The finding on systemic clearance times could be important - studies in Europe where Anthelmintic is POM V/ strict policies on use and there is some re-emergance of large strongyles appearing (Denmark and Sweden)

The findings need to have the limitations highlighted e.g. sample size and how this might influence findings with the caveat that even with these limitations these data still provide a development on athelmintic efficacy and activity for IVM in mules. 

Other minor comments:

Line 44 spelling of remained

Line 168 The not efficacy?

Line 208 LM do you mean ML - this is also used later in the manuscript

line 210 needs "have" inserting

Line 211 - remove "also"

Line 215 donkey or donkeys ?

Lines 225 and 241 - this is a statement not a paragraph

With revisions a good study with valuable contribution to the field.

Author Response

The discussion section requires some work in the English language and also further clarity and highlighting limitation which currently are lacking. The discussion and conclusion were both modified according with the reviewer’s remarks.

To clarify were all the mules on exactly the same management and diet? All the enrolled animals were undergoing to the same management and diet; this information was added in the M&M

Why were only females used for blood sampling and why only 5? Is there rationale for this? What effect might this have on statistical power? Worth a limitation comment in the discussion. The mules were chosen on the basis of a quieter temperament, less resistance and ease of handling, and they happened to all be females. This information has been added in the manuscript.  Five animals were considered appropriate as sample size because a similar numerosity was used in other studies relating to IVM pharmacokinetics in horse and donkey following oral administration at the same dosage. Please, see the Table 2 in which we have provided to insert the sample size used in the above-cited studies.

Similarly why 15 mules how might the sample size reflect findings and how comparable is this to the other horse studies cited? The aim of the present work was to assess the efficacy of IVM in mule using a FECR method that represents the gold standard method for in vivo efficacy field trial; specific guidelines for FECRT in horses are being developed by parasitologists under the auspices of the World Association for the Advancement of Veterinary Parasitology (WAAVP); these guidelines declare that the ideal group size to perform a FECRT is around 10 to 15 animals but for horses also six animals on each farm may be considered a correct number. Since FECs are, by their very nature quite variable, and testing just few horses could lead to an incorrect inference, we decided to use 15 animals that is a very good number for this tool.  

Line 232 - comment on suppression of egg laying activity - do you have a citation for this - we have seen it in horses for Benzimidazoles. Thank you for the comment, maybe the sentence was not clear, we did not talk about the Eggs Laying Test to assess anthelmintic resistance in vitro against benzoimidazoles in GI strongyles of small ruminants, we have modified the sentence.

Line 235- comment on ERP - ERP was calculated at 28 days but not further. The guidelines used here AAEP are good but in the literature there are multiple ways to calculate ERP e.g. Relf et al 2014 and Daniels & Proudman 2016 - both used 90% reduction post 14 days as reduced efficacy for which you did not observe, therefore you observed good efficacy in this study. The FECs after a treatment can be used to assess if strongyles are resistant to a given anthelmintic and also to test the efficacy of the product as well, that represents our primary goal. At the beginning we did not think to use FEC to assess ERP but only FECR at +14 days post treatment for efficacy, than we added an other FEC at +28 days because it was in line with the last blood sampling and could give additional information on the rate of efficacy. In fact ML are characterized by very long ERPs, but recent reports have documented them shortened to just 4-5 weeks that could represent a first signal of suspected resistance. Since at days +14 we have detected a rate of efficacy of IVM lower than expected, we decided to test again the animals to assess an eventually higher decrease of the efficacy that could be attributable to suspect anthelmintic resistance. Not additional evaluation were possible because of the bad temperament of the animals and also for the lack of the approval by the Ministry of Health for sampling longer than 30 days.

It might be of value to comment on the encysted burden? This is possible to test via serum from Austin Davies Diagnostics - could this have any influence on findings? We have included a sentence about possible lack of efficacy on the encysted parasites (L3-L4) by IVM that could have impact on the FEC post-treatment; the ELISA conducted on serum to assess encysted parasites in general can be used at the beginning of spring in clinical field when a larval cyathostominosis is suspected in absence of a coprological confirmation. Honestly we don’t have experiences on the useful of antibody detection in the interpretation of drug efficacy. Morover, since antibodies need some months to be catabolized, the antibody detected by serological assays may not be necessarily expression of the encysted parasitic load at the determination time, but may represent an expression of a past infection.

The findings need to have the limitations highlighted e.g. sample size and how this might influence findings with the caveat that even with these limitations these data still provide a development on athelmintic efficacy and activity for IVM in mules. Consideration are included in the conclusions.

Other minor comments:

Line 44 spelling of remained. Change was done.

Line 168 The not efficacy?. Change was done

Line 208 LM do you mean ML - this is also used later in the manuscript. Change was done.

line 210 needs "have" inserting. Change was done.

Line 211 - remove "also". Change was done.

Line 215 donkey or donkeys ? Changes were done.

Reviewer 2 Report

Manuscript ID: animals-783446

Comments to Author

Overview and Major Comments:

The manuscript reports on the anthelmintic efficacy and pharmacokinetics of ivermectin paste after oral administration in mules. Mules with confirmed cyathostomin infestations (> 200epg) were included in the study. The dose of ivermectin used was that registered for use in horses (200µg/kg BW). The pharmacokinetic profile of ivermectin was investigated using appropriate parameters that are frequently used in pharmacokinetic studies eg. Cmax, Tmax, AUC. Faecal egg count reduction tests were used to determine the efficacy of ivermectin and this is commonly used for this purpose. There is very little known about the pharmacokinetic differences between horses and mules and doses of many drugs used in mules are extrapolated from recommended doses in horses. Incorrect dosing of anthelmintics (namely underdosing) can lead to reduced efficacy and hence, contribute to the development of anthelmintic resistance. This research is therefore useful to ensure appropriate doses of ivermectin are recommended and used in mules.

Overall, the paper is quite well written, however there are multiple mistakes with English language/style that need to be corrected prior to publication.

More information about the pharmacokinetics of ivermectin should be discussed in the introduction ie. the authors should provide a summary of how the drug is absorbed and eliminated from the body.

15 mules were included in the study and although a power calculation is not provided, similar pharmacokinetic studies in horses often use a lower number of horses. By including this number of mules in the study, variation in pharmacokinetic variables among individual animals may be more easily identified and have less impact on the overall results of the study.

The methods used for faecal egg counts, faecal egg count reduction tests and coproculture are appropriate and well described. However, it is assumed only a single egg count was performed on a single faecal sample from each mule enrolled in the study at each time point. The limitations associated with variability in faecal egg counts is not discussed as a possible limitation in this study and is also not addressed ie. performing more than one count may have provided a better estimate of the mean egg count distribution in each mule. The variability in FEC results can also lead to inaccuracy in FECR results and subsequently inaccurate conclusions regarding anthelmintic efficacy and/or the presence of resistance. This should also have been discussed as a potential limitation of the study.

The sampling for the pharmacokinetics component of the study involved collection of whole blood, with analysis performed on serum. Previous studies in horses and donkeys investigating the pharmacokinetics of ivermectin have used plasma samples. There is no clear reason given in this paper for why serum was used. This is then contradicted in the results section 3.2 which describes the ivermectin concentration in plasma.

The results are presented clearly, however only the arithmetic mean of the egg counts are included. The raw egg count data should be provided as supplementary information. The FECR results are also difficult to interpret. The FECR increases from day 14 and day 28 and this is not what would be expected. Although ivermectin is eliminated slowly from body compartments, the level of egg suppression would be expected to be higher at 14 days post treatment. Possible reasons for why there was an increased percentage in egg count reduction at 28 days need to be included in the discussion. It is also not clearly stated or justified why a FECR was performed at 14 and 28 days post-treatment, when this test is usually only performed at 14 days post-treatment.

The opening sentence and paragraph of the discussion should be rewritten and should summarise the conclusions made from this study instead of discussing general literature regarding ivermectin and its use in horses.

Some of the conclusions drawn in the discussion are not discussed in enough detail and are not always sufficiently supported by the results. A FECR result of 96.57% at 14 days, as in this study, should be interpreted as suspected resistance to ivermectin and inadequate efficacy of the drug (as outlined in the AAEP guidelines for equine parasite control). This is not well discussed in this paper and the authors suggest that it is inconclusive evidence of resistance. This statement is not well justified. It is possible and likely that resistance in cyathostomins in mules to ivermectin is developing and increasing at a similar rate as has been described in horses as supported by equivalent FECR results in horses in previous studies. This needs to be discussed in more detail.

The authors state that there are several reasons why there may be decreased efficacy of ivermectin, however they then only give one reason ie. underdosing.

Although there is an increase in FECR by 28 days and this could be interpreted as adequate efficacy of ivermectin, this should be discussed further including justification for why FECR was performed at 28 days and potential reasons for this increase ie. the inherent variability in FECR results may decrease the accuracy of FECR results.

There is insufficient discussion regarding the results of the pharmacokinetic investigations in this study. The authors compare the results to those in horses and donkeys but there is no further discussion as to whether the results in mules are suggestive that the dose used is appropriate which is one of the main aims of the study.

The authors discuss the potential impact of the level of parasitism on the pharmacokinetic results, however the comparison with the results found in lambs is not applicable as this was following  subcutaneous administration of ivermectin and no significant difference was observed between parasitized and non-parasitized lambs after oral administration of ivermectin.

The authors conclude that ivermectin is efficacious against cyathostomins in mules but the results of this study do not adequately support this statement.

Specific Comments:

Line 18: The wording is inappropriate in this sentence – the phrase ‘under clinical and therapeutic points of view does not make sense’.

Line 19: This sentence should also be reworded. The second half of the sentence doesn’t make sense. What do the authors mean by ‘with reflex on the pharmacological effects’?

Line 23: please reword this sentence. It should also be drug resistance (not resistances).

Line 27: please reword this sentence – ‘Pharmacokinetic behaviour in mules showed intermediate responses found in between….’ needs to be rephrased.

Line 28: Please reword this sentence. It currently reads that ‘Studies aimed to…..could control parasites and improve welfare’ and this needs to be phrased differently.

Line 31: Change from ‘ivermectin represents’ to ‘ivermectin is’ and reword ‘thanks to the….’

Line 32: Change from ‘use of IVM in the mule at the same conditions’ to ‘under the same conditions’.

Line 34: Change to ‘The aim’ and ‘infected with cyathostomins’.

Line 39: Please reword this sentence eg. ‘FECs were performed before and after treatment at days 14 and 28, using a modified McMaster method. The percentage FEC reduction (FECR%) was also calculated.’

Line 44: ‘remained’ is spelt incorrectly.

Line 50: Change to ‘there has been a decline in mule populations…’

Line 53: Change to ‘there has been an increase in the use of mules…’. Please reword the second half of the sentence.

Line 56: Change to ‘many authors consider’

Line 57 and 59: Avoid using ‘pathogens’ to describe parasites. ‘Nematodes’ should be used instead.

Line 59: Please reword this sentence. The term ‘fitness’ is not really appropriate here.

Line 68: Please reword the second part of this sentence ie. ‘commonly used thanks to….’. Please also change ‘endo’ to ‘endo-‘

Line 70: Please check the English used in this sentence

Line 75: This sentence is too long and please check the English used.

Line 81: What do the authors mean by ‘In this context…’?

Line 97: remove ‘of the’ towards the end of the sentence.

Line 99: Please change to ‘All mules used in the study had a FEC ≥200 eggs per gram, with exclusive presence of cyathostomins’.

Line 143: Please change to ‘The HLPC equipment consisted of…’

Line 151: Please change to ‘a series of five concentration points prepared by spiking blank serum samples were injected for calibration’.

Line 153: Change to ‘in accordance with’ EMA guidelines’.

Line 156: Change to ‘the upper limit’

Line 159: Please check the English used in this sentence

Line 161: This sentence needs to be reworded. The efficacy of IVM cannot be determined at each fecal sampling time.

Line 168: Please check the English used in this sentence. ‘Not efficacy’ is not appropriate terminology.

Line 177: Please change to ‘consisting of 100% cyathostomins’

Line 180: Please check this Table legend for errors in the wording and English language

Line 208: Please change to ML. There are additional occurrences of this error in the remainder of the discussion.

Line 209: Please check the English used in this sentence.

Line 215: This sentence should be changed to ‘Pharmacokinetic differences may alter the efficacy of a drug…’

Line 216: The structure/order of this sentence should be changed eg. ‘Latzel et al. (2012) observed a significant difference in the half-life of xylazine (greater in horses than in mules) following IV administration and at the same time, a less intense sedative effect in mules.’

Line 221: This sentence would be better placed in the introduction

Line 225: Change to ‘orally administered..’

Line 227: This sentence needs to be reworded. What do the authors mean by ‘incomplete’ efficacy?

Line 229: This sentence needs to be reworded.

Please 234: Please check the English used in this sentence.

Line 235: Please check the English used in this sentence.

Line 241: This sentence is not well placed and does not make sense. The authors mention ‘extra intestinal migration’ – are they instead referring to the hypobiosis that can occur with cyathsotomins?

Line 249: Please check the English used in this sentence.

Line 265: Please check the English used in this sentence.

Line 268: Please check the English used in this sentence.

Line 273: Please check the English used in this sentence.

Author Response

More information about the pharmacokinetics of ivermectin should be discussed in the introduction ie. the authors should provide a summary of how the drug is absorbed and eliminated from the body.

Thank you for your suggestion, some information about pharmacokinetics of IVM in domestic animal in general and in horse have been added.

15 mules were included in the study and although a power calculation is not provided, similar pharmacokinetic studies in horses often use a lower number of horses. By including this number of mules in the study, variation in pharmacokinetic variables among individual animals may be more easily identified and have less impact on the overall results of the study. As suggested from reviewer 1, we have added in M&M that 5 out of the 15 mules are included in the study on the basis of their more quiet temperament.

The methods used for faecal egg counts, faecal egg count reduction tests and coproculture are appropriate and well described. However, it is assumed only a single egg count was performed on a single faecal sample from each mule enrolled in the study at each time point. The limitations associated with variability in faecal egg counts is not discussed as a possible limitation in this study and is also not addressed ie. performing more than one count may have provided a better estimate of the mean egg count distribution in each mule. The variability in FEC results can also lead to inaccuracy in FECR results and subsequently inaccurate conclusions regarding anthelmintic efficacy and/or the presence of resistance. This should also have been discussed as a potential limitation of the study.  The FEC determination at D+14 and D+28 were based on individual faecal sample as suggested by the WAAVP, the possible variation in the individual strongyle egg excretion are minimized by the use of the average FEC used for the FECR; some Authors analyses longitudinal fecal egg count (FEC) data on horse at individual level showing that there is a consistency in FECs over a time that is generally weak and accounted for <10% of the total variance but the variation in excretions was observed over weeks or months and not within the same day, in a recent study Helena Carstensen et al. (Journal of Equine Veterinary Science 33 (2013) 161-164) conclude that there is no significant within-day variability in the shedding of strongyle eggs in horses, and the time of day for collection of fecal samples does not represent a source of error in research studies.

The sampling for the pharmacokinetics component of the study involved collection of whole blood, with analysis performed on serum. Previous studies in horses and donkeys investigating the pharmacokinetics of ivermectin have used plasma samples. There is no clear reason given in this paper for why serum was used. This is then contradicted in the results section 3.2 which describes the ivermectin concentration in plasma. Thank you for your observation about section 3.2. The term plasma was used instead of serum for a mistake; now we have provided to correct it. We agree that in previous studies the plasma was used. Generally, plasma is chosen because with it is possible to obtain a major volume to analyze, but the anticoagulant could be interfered with the assay. We have token about 10mL of blood at each sampling time, for analytical determination was used only 1 mL, as a consequence we had serum for more replicates when necessary.  The analytical method was validated according EMA guidelines on bioanalytical methods and perfectly able to determine IVM concentrations in systemic circulation.

The results are presented clearly, however only the arithmetic mean of the egg counts are included. The raw egg count data should be provided as supplementary information. -Data are given as supplementary data. The FECR results are also difficult to interpret. The FECR increases from day 14 and day 28 and this is not what would be expected. Although ivermectin is eliminated slowly from body compartments, the level of egg suppression would be expected to be higher at 14 days post treatment. Possible reasons for why there was an increased percentage in egg count reduction at 28 days need to be included in the discussion. It is also not clearly stated or justified why a FECR was performed at 14 and 28 days post-treatment, when this test is usually only performed at 14 days post-treatment. The average FECs obtained at the two distinct sampling times did not differ from a statistical point of view (supplementary data added), therefore the little variation on FECs that effects on the FECR% may be introduced both by an individual variation of excretion quite common in horses (and may be also in mule), but also by the analytic sensitivities the Mc Master technique, which is still considered the reference method for the FECR% tests. We added a sentence about this in the discussion to better clarify.

The opening sentence and paragraph of the discussion should be rewritten and should summarise the conclusions made from this study instead of discussing general literature regarding ivermectin and its use in horses. Change was done.

Some of the conclusions drawn in the discussion are not discussed in enough detail and are not always sufficiently supported by the results. A FECR result of 96.57% at 14 days, as in this study, should be interpreted as suspected resistance to ivermectin and inadequate efficacy of the drug (as outlined in the AAEP guidelines for equine parasite control). This is not well discussed in this paper and the authors suggest that it is inconclusive evidence of resistance. This statement is not well justified. It is possible and likely that resistance in cyathostomins in mules to ivermectin is developing and increasing at a similar rate as has been described in horses as supported by equivalent FECR results in horses in previous studies. This needs to be discussed in more detail.

The authors state that there are several reasons why there may be decreased efficacy of ivermectin, however they then only give one reason ie. underdosing.  As suggested by the reviewer additional comments on the reduced efficacy were added on the conclusions even if it should always be borne in mind that a borderline reduced efficacy, as it is, can be caused by factors other than resistance for instance the issues of the under dosing that needs of course further investigation. A failure to administer the recommended appropriate dose to the animal being treated can be incorrectly interpreted as resistance, can be excluded in this case because was done from the Authors. We think resistance is extremely unlikely and that resistant worms are not present in the sampled mules for a number of reason: a) all the mules sampled did not share pastures, and thus parasite populations, with other animals e.g horses or donkeys, that could introduce resistance across that population; b) as written in the materials and methods all the herd population come from an internal mating and any external introduction of animals, harboured resistant strains, was done in the last 10 years; c) pharmacological pressure within the herd was minimal since the frequency of anthelmintic treatments was very low and the deworming was occasional.

Although there is an increase in FECR by 28 days and this could be interpreted as adequate efficacy of ivermectin, this should be discussed further including justification for why FECR was performed at 28 days and potential reasons for this increase ie. the inherent variability in FECR results may decrease the accuracy of FECR results. Comments were included in the discussion.

There is insufficient discussion regarding the results of the pharmacokinetic investigations in this study. The authors compare the results to those in horses and donkeys but there is no further discussion as to whether the results in mules are suggestive that the dose used is appropriate which is one of the main aims of the study. Thank you, you are correct. Now, we have added a comment about limits and utility of PK results in the therapy of mules.

The authors discuss the potential impact of the level of parasitism on the pharmacokinetic results, however the comparison with the results found in lambs is not applicable as this was following  subcutaneous administration of ivermectin and no significant difference was observed between parasitized and non-parasitized lambs after oral administration of ivermectin. We agree with you. The sentence was deleted.

Specific Comments:

Line 18: The wording is inappropriate in this sentence – the phrase ‘under clinical and therapeutic points of view does not make sense’. Change was done.

Line 19: This sentence should also be reworded. The second half of the sentence doesn’t make sense. What do the authors mean by ‘with reflex on the pharmacological effects’? Change was done.

Line 23: please reword this sentence. It should also be drug resistance (not resistances). Change was done.

Line 27: please reword this sentence – ‘Pharmacokinetic behaviour in mules showed intermediate responses found in between….’ needs to be rephrased. Change was done.

Line 28: Please reword this sentence. It currently reads that ‘Studies aimed to…..could control parasites and improve welfare’ and this needs to be phrased differently. Change was done.

Line 31: Change from ‘ivermectin represents’ to ‘ivermectin is’ and reword ‘thanks to the….’ Change was done.

Line 32: Change from ‘use of IVM in the mule at the same conditions’ to ‘under the same conditions’. Change was done.

Line 34: Change to ‘The aim’ and ‘infected with cyathostomins’. Change was done.

Line 39: Please reword this sentence eg. ‘FECs were performed before and after treatment at days 14 and 28, using a modified McMaster method. The percentage FEC reduction (FECR%) was also calculated.’ Change was done.

Line 44: ‘remained’ is spelt incorrectly. Change was done.

Line 50: Change to ‘there has been a decline in mule populations…’ Change was done.

Line 53: Change to ‘there has been an increase in the use of mules…’. Please reword the second half of the sentence. Change was done.

Line 56: Change to ‘many authors consider’ Change was done.

Line 57 and 59: Avoid using ‘pathogens’ to describe parasites. ‘Nematodes’ should be used instead. Change was done.

Line 59: Please reword this sentence. The term ‘fitness’ is not really appropriate here.

Line 68: Please reword the second part of this sentence ie. ‘commonly used thanks to….’. Please also change ‘endo’ to ‘endo-‘ Change was done.

Line 70: Please check the English used in this sentence. Change was done.

Line 75: This sentence is too long and please check the English used. Change was done.

Line 81: What do the authors mean by ‘In this context…’? Change was done.

Line 97: remove ‘of the’ towards the end of the sentence. Change was done.

Line 99: Please change to ‘All mules used in the study had a FEC ≥200 eggs per gram, with exclusive presence of cyathostomins’. Change was done.

Line 143: Please change to ‘The HLPC equipment consisted of…’ Change was done.

Line 151: Please change to ‘a series of five concentration points prepared by spiking blank serum samples were injected for calibration’. Change was done.

Line 153: Change to ‘in accordance with’ EMA guidelines’. Change was done.

Line 156: Change to ‘the upper limit’ Change was done.

Line 159: Please check the English used in this sentence. Change was done.

Line 168: Please check the English used in this sentence. ‘Not efficacy’ is not appropriate terminology. Change was done.

Line 177: Please change to ‘consisting of 100% cyathostomins’ Change was done.

Line 208: Please change to ML. There are additional occurrences of this error in the remainder of the discussion. Change was done.

Line 209: Please check the English used in this sentence. Change was done.

Line 215: This sentence should be changed to ‘Pharmacokinetic differences may alter the efficacy of a drug…’ Change was done.

Line 216: The structure/order of this sentence should be changed eg. ‘Latzel et al. (2012) observed a significant difference in the half-life of xylazine (greater in horses than in mules) following IV administration and at the same time, a less intense sedative effect in mules.’ Change was done.

Line 221: This sentence would be better placed in the introduction. Change was done.

Line 225: Change to ‘orally administered..’ Change was done and the sentence was moved at the start of manuscript.

Line 227: This sentence needs to be reworded. What do the authors mean by ‘incomplete’ efficacy? Change was done.

Line 229: This sentence needs to be reworded. Change was done.

Please 234: Please check the English used in this sentence. Change was done.

Line 235: Please check the English used in this sentence. Change was done.

Line 249: Please check the English used in this sentence. Change was done.

Line 265: Please check the English used in this sentence. Change was done.

Line 268: Please check the English used in this sentence. Change was done.

Line 273: Please check the English used in this sentence. Change was done.

Round 2

Reviewer 2 Report

The revised version of the manuscript has adequately addressed the changes suggested, however some editing to ensure appropriate use of English language is still required.